# Missense Variants in *GFRA1* and *NPNT* Are Associated with Congenital Anomalies of the Kidney and Urinary Tract

**DOI:** 10.3390/genes13101687

**Published:** 2022-09-21

**Authors:** Mohamed H. Al-Hamed, John A. Sayer, Nada Alsahan, Noel Edwards, Wafaa Ali, Maha Tulbah, Faiqa Imtiaz

**Affiliations:** 1Department of Clinical Genomics, Center for Genomic Medicine, King Faisal Specialist Hospital and Research Center, Riyadh 11211, Saudi Arabia; 2College of Medicine, Alfaisal University, Riyadh 11533, Saudi Arabia; 3Translational and Clinical Research Institute, Faculty of Medical Sciences, Newcastle University, Central Parkway, Newcastle upon Tyne NE1 3BZ, UK; 4Renal Services, The Newcastle upon Tyne Hospitals NHS Foundation Trust, Newcastle upon Tyne NE7 7DN, UK; 5Department of Obstetrics and Genecology, King Faisal Specialist Hospital and Research Center, Riyadh 11211, Saudi Arabia

**Keywords:** CAKUT, renal agenesis, *GFRA1*, *NPNT*, prenatal exome

## Abstract

The use of next-generation sequencing (NGS) has helped in identifying many genes that cause congenital anomalies of the kidney and urinary tract (CAKUT). Bilateral renal agenesis (BRA) is the most severe presentation of CAKUT, and its association with autosomal recessively inherited genes is expanding. Highly consanguineous populations can impact the detection of recessively inherited genes. Here, we report two families harboring homozygous missense variants in recently described genes, *NPNT* and *GFRA1*. Two consanguineous families with neonatal death due to CAKUT were investigated. Fetal ultrasound of probands identified BRA in the first family and severe renal cystic dysplasia in the second family. Exome sequencing coupled with homozygosity mapping was performed, and Sanger sequencing was used to confirm segregation of alleles in both families. In the first family with BRA, we identified a homozygous missense variant in *GFRA1*: c.362A>G; p.(Tyr121Cys), which is predicted to damage the protein structure. In the second family with renal cystic dysplasia, we identified a homozygous missense variant in *NPNT*: c.56C>G; p.(Ala19Gly), which is predicted to disrupt the signal peptide site. We report two Saudi Arabian consanguineous families with CAKUT phenotypes that included renal agenesis caused by missense variants in *GFRA1* and *NPNT*, confirming the role of these two genes in human kidney development.

## 1. Introduction

Congenital anomalies of the kidney and urinary tract (CAKUT) are a spectrum of abnormalities affecting the morphogenesis of the kidneys or other structures of the urinary tract. CAKUT is one of the most common congenital defects affecting 3–7 out of 1000 live births [1]. Bilateral renal agenesis (BRA) is the most severe presentation of CAKUT. 

BRA is an example of a fatal neonatal kidney disease. The failure of both kidneys to develop in utero results in oligohydramnios, and lack of amniotic fluid may cause compression of the fetus and further fetal malformations. The frequency of BRA is approximately 1/3000–1/5000 births [2], while unilateral renal agenesis (URA) is more common at 1/1000–2000 [3] and is usually clinically silent.

BRA is more common in infants with a parent who has a renal anomaly, particularly unilateral renal agenesis. Studies have shown that URA and BRA may be genetically related [4]. In humans, the kidneys develop between the 5th and 14th week of fetal development, and by the 14th week, they are normally producing urine [5]. Approximately 40% of fetuses with bilateral renal agenesis will be stillborn, and if born alive, the baby will usually live only a few hours [6].

Currently, the precise genetic causes of CAKUT are not known. In recent years, alterations in more than 75 genes have been shown to cause isolated or syndromic CAKUT [7], where an autosomal dominant mode of inheritance is more frequent than an autosomal recessive pattern of inheritance. Classically, mouse models of CAKUT have helped to identify many genes associated with human CAKUT phenotypes, and recently, with the advent of next-generation sequencing (NGS), many more genes associated with CAKUT have been identified [8].

The *NPNT* gene on chromosome 4q25 by [9] consists of 13 exons and encodes for a protein of 561 amino acids. The NPNT protein, also named nephronectin, is expressed in the fetal cochlea, eye, heart, lung, and embryonic kidney cells [9]. The NPNT protein is an extracellular matrix protein localized at the glomerular basement membrane (GBM) [10] and associated with other epithelial structures (Wolffian duct and ureteric bud) that have well-defined roles in kidney development [11]. A knockdown of *NPNT* leads to podocyte dysfunction and GBM disorganization [12]. A recent report by Dai and colleagues highlighted the role of the loss of function of an *NPNT* variant and BRA in a consanguineous Chinese family [13]. We hypothesize that as well as the reported frameshift variants, homozygous missense variants in *NPNT* that are predicted to affect protein structure and function may also lead to severe CAKUT phenotypes, including BRA.

The *GFRA1* gene was mapped to chromosome 10q25.3 and comprises nine exons that encodes a 465-amino-acid polypeptide receptor [14]. It is an important receptor for glial-cell-line-derived neurotrophic factor (GDNF) protein [15]. The complex signaling of GFRA1/GDNF and RET protein-tyrosine kinase is critical to the development of the kidney [15]. Recently, biallelic loss-of-function variants in the *GFRA1* gene were reported in three unrelated consanguineous families with BRA by two groups [16,17]. Here, we hypothesize that as well as the reported nonsense variants. Homozygous missense variants in *GFRA1* that affect protein structure and function may also lead to CAKUT phenotypes.

Here, we report two consanguineous families with fetuses affected in an autosomal recessive pattern with variable features of CAKUT. In the first family, we identified a homozygous missense variant in *GFRA1*, and in the second family, we identified a homozygous missense variant in *NPNT*. This is the first report to associate homozygous missense variants in *GFRA1* and *NPNT* with CAKUT phenotypes.

## 2. Materials and Methods

### 2.1. Human Subjects

Two families presented to the Maternal Fetal Medicine High-Risk Clinic at the King Faisal Specialist Hospital and Research Centre (KFSH&RC) because of the recurrence of pregnancy loss in their offspring due to congenital renal anomalies and were subsequently recruited following informed and written consent. The study was approved by the Research Advisory Council at the King Faisal Specialist Hospital and Research Centre (KFSH&RC), Riyadh, Saudi Arabia (RAC# 2160 022). Fetal DNA was extracted from cord blood. Parental and live sibling DNA was extracted using a peripheral blood sample.

### 2.2. Homozygosity Mapping and Trio-Exome Sequencing

Using genomic DNA from affected fetuses, chromosomal microarray (CMA) testing was performed using the Affymetrix CytoScan assay platform according the manufacturer’s instructions. Given the known consanguinity, regions of homozygosity (ROH) > 2 Mb were used as surrogates of autozygosity to search for autosomal recessive causes of disease. Exome sequencing (ES), a technique for sequencing all of the protein-coding regions of the genome, using the Illumina HiSeq 2500 platform and TruSeq DNA exome capture with a ≥98% coverage of RefSeq and a >85% coverage of 20× read depth was performed. Downstream data analysis and subsequent filtering of variants by CAKUT candidate gene coordinates was performed in both families. Sanger sequencing validation was performed for identified candidate variants within homozygous regions. Oligonucleotide primers for PCR amplification of targeted variants were designed using Primer3 software (http://frodo.wi.mit.edu/ accessed on 17 May 2022) and synthesized in-house. The amplified PCR products were sequenced using an ABI 3730xl capillary sequencer (Applied Biosystems, Foster City, CA, USA), and sequences were analyzed using Mutation Surveyor software V.3.24 (SoftGenetics LLC, State College, PA, USA).

### 2.3. In Silico Protein Modeling 

The protein domains of human NPNT (accession number NP_001028219.1) and GFRA1 (accession number NP_001335027.1) were modelled using in silico tools. AlphaFold2 [18] was utilized to predict the three-dimensional structure of the N-terminal signal peptide domain, and figures were prepared using PyMOL (http://www.pymol.org/ accessed on 23 May 2022). SignalP 5.03 [19] was used to the predict cleavage site with a signal peptide domain.

## 3. Results

### 3.1. Family 1 with Bilateral Renal Agenesis

A consanguineous family who had lost three pregnancies with three neonatal deaths due to BRA and five healthy children was investigated (Figure 1). The family history confirmed healthy parents and a neonatal death in the first, second, and fifth pregnancies. This is consistent with an autosomal recessive cause of disease in this family. In the fifth pregnancy, fetal antenatal ultrasound indicated BRA and anhydramnios at 20 weeks’ gestation. Cordocentesis was performed to obtain fetal DNA for the CMA, which is a first-tier diagnostic test and allows the testing of chromosomal imbalances, duplications, and deletions, and showed no abnormalities. Genomic DNA from peripheral blood samples was obtained from all available family members (including both parents and five asymptomatic children) for homozygosity mapping and trio-ES. Homozygosity mapping showed a region of homozygosity on chromosome 10. ES detected a novel homozygous missense variant in *GFRA1* (NM_001348098.4) c.362A>G; p.(Tyr121Cys), which was confirmed by Sanger sequencing (Figure 1). Segregation analysis revealed that the parents and three unaffected siblings were heterozygous for the allele, and two unaffected siblings were wild type for the allele. The *GFRA1* variant was not found in public databases (gnomAD, 1000 Genomes Project, ESP, and the Saudi Arabian Center for Genomic Medicine (CGM-DB)), and in silico prediction tools and conservation analysis predicted that this variant was probably damaging to the protein structure and function (Figure 1). The linker domain containing 119SPYE122 is found in GFRα1–GFRα3 (but not GFR4 or GFRAL), is highly conserved, and allows domains 1 and 3 of GFRα1 to pack against each other [20]. The Tyr121Cys variant is within this linker, and the changes in the physiochemical properties of the side chain at position 121 may disrupt the overall structural organization of GFRα1 pointing to a potential mechanism of pathogenicity.

### 3.2. Family 2 with Renal Cystic Dysplasia

A first-degree consanguineous family had two unaffected children the first born by cesarean section due to pre-eclampsia and the third born at full term by vaginal delivery (Figure 2). The second pregnancy (proband) was referred at 31 weeks’ gestation, where antenatal ultrasound findings indicated single viable fetus with hyperechoic kidneys bilaterally indicating severe cystic dysplasia. There was also anhydramnios with nonvisualized bladder and stomach. Cordocentesis was performed, and extracted DNA was used for CMA and trio-ES sequencing together with parental DNA. At 33 weeks’ gestation, the fetus was diagnosed with intrauterine fetal death and underwent a spontaneous vaginal delivery. CMA results showed no abnormalities while trio-ES (using fetal and parental DNA) revealed a homozygous missense variant in the *NPNT* gene (NM_001033047.3: c.56C>G; (p.Ala19Gly). Subsequent genetic testing of the third (unaffected) child, using Sanger sequencing, revealed that she was heterozygous for the allele (Figure 2).

The missense variant in *NPNT* is predicted to be deleterious by disrupting the signal peptide site at amino acid residue 19 of the NPNT protein. In addition, cross-species conservation analysis shows that the affected residue is strongly conserved down to *Danio rerio*. The allele frequency of the missense variant *NPNT*, c. 56C>G (rs1265091172), is 0.00001070 in gnomAD and 0.000420 in the Center for Genomic Medicine (CGM-DB) and has not been seen homozygously. The alteration leads to a substitution of conserved glycine residue by alanine at the last amino acid of the N-terminal signal region of the NPNT protein. The AlphaFold2 protein modelling software predicted a three-dimensional structure of the N-terminal signal peptide domain and predicted a cleavage site between Ala19 and Glu20 (Figure 2). Furthermore, in silico prediction tools predicted a deleterious effect of the alteration, as shown in Table 1. These in silico measures provide strong supportive evidence of pathogenicity of the *NPNT* allele.

## 4. Discussion

The disruption of normal nephrogenesis due to environmental and genetic causes is the basis of CAKUT pathogenesis [1]. BRA is lethal and represents the severe form of the CAKUT. Syndromic BRA associated with multiple congenital malformation is more common than isolated BRA. There are a number of genes with autosomal recessive inheritance reported to be associated with isolated BRA; these include *ITGA8* [21], *FGF20* [22], *GFRA1*, [16] and *NPNT* [16]. The reports for the most recent genes *NPNT* [13] and *GFRA1* [16] indicated biallelic loss of function variants in association with BRA. Here, we report the association of homozygous missense variants in *GFRA1* and *NPNT* with CAKUT. The *GFRA1* missense variant c.362A>G was associated with BRA, while the missense variant in *NPNT*: c.56C>G was associated with severe renal cystic dysplasia. In silico analysis suggested that the variant in *GFRA1* may disrupt the overall structural organization of GFRA1 and, hence, reduce GDNF-GFRA1-RET signaling. The previous cases reported with *GFRA1* variants are from the United Arab Emirates and Oman, both located near Saudi Arabia. 

The *NPNT* missense variant we identified was associated with severe renal cystic dysplasia but not with BRA or URA, as reported previously [13,23]. Although the pathogenesis of the missense variant is less clear, we note that in silico modeling suggests a deleterious effect on the signal peptide. The variability of CAKUT as a phenotype is well established, and we postulate that variants in *NPNT* are no exception despite the clear tendency towards a lethal phenotype. In a similar way, it has been reported previously [24] that missense variants in the Fraser/MOTA/BNAR spectrum genes cause milder CAKUT in comparison with truncating mutations that lead to a severe form of Fraser syndrome. 

This study has some limitations. For the novel alleles in *GFRA1* and *NPNT*, we have identified just one family each, both from highly consanguineous pedigrees. However, these families combined with those already reported expand both the genotypic and phenotypic spectrum. It allows the rare variants *GFRA1* and *NPNT* to be considered as a cause of BRA or CAKUT and suggests that the structural modelling of missense alleles is helpful to contribute towards the overall pathogenicity, given the often-seen discrepancies using in silico tools, such as PolyPhen-2 and SIFT. 

## 5. Conclusions

In this report, we detail two consanguineous families with fetuses affected in an autosomal recessive pattern with variable features of CAKUT from renal agenesis to cystic renal dysplasia. The first family a *GFRA1* homozygous missense allele was identified, and in the second family, an *NPNT* homozygous missense allele was found. This is the first report to associate homozygous missense variants in *GFRA1* and *NPNT* with CAKUT phenotypes. We conclude that *GFRA1* and *NPNT* are bono fide CAKUT genes, and we extend both the genotypic and phenotypic spectrum of these genes.

## Figures and Tables

**Figure 1 genes-13-01687-f001:**
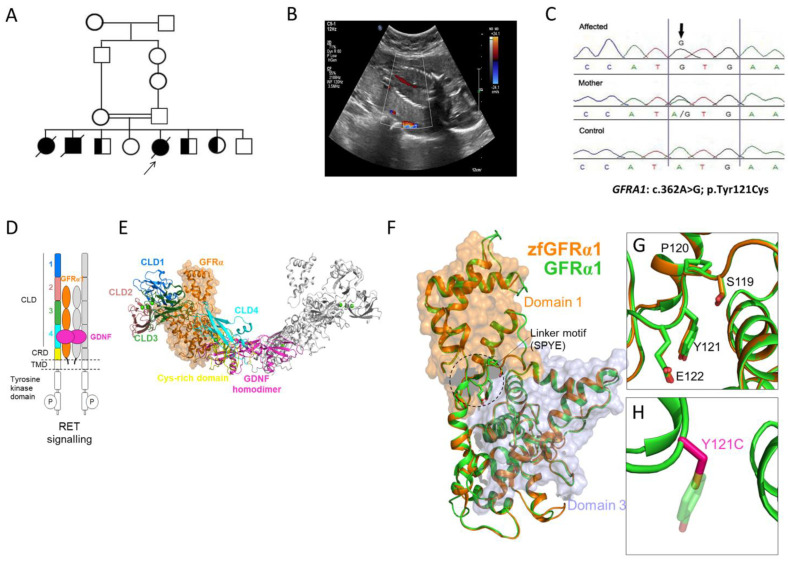
**Family 1 with bilateral renal agenesis and *GFRA1* variant.** (**A**) Pedigree diagram of the studied family with bilateral renal agenesis with the proband arrowed. (**B**) Coronal view of the fetal abdomen at 20 weeks’ gestation with color Doppler showing the abdominal aorta with no evidence of the left or right renal artery branching, indicating bilateral renal agenesis. (**C**) Sanger sequencing chromatogram of the GFRA1 missense variant (NM_001348098.4: c.362A>G; p.Tyr121Cys) in the affected proband, mother, and control. (**D**) Schematic of the GDNF-GFRα1-RET signaling system. Receptor tyrosine kinase (RET) dimerization and signaling is activated by association with GDNF (magenta) bound to its GFRα1 co-receptor (orange). For clarity, one-half of the dimerized signaling complex is highlighted: extracellular RET cadherin-like domains (CLD) 1 (blue), 2 (pink), 3 (green), and 4 (cyan) and cysteine-rich domain (CRD) are shown in yellow. (**E**) The cryo-EM structure (colored as in part D) of the GDNF-GFRα1-RET zebrafish (zf) assembly (PDB: 7AML). (**F**) The cryo-EM structure of the zfGFRα1 (orange) and human GFRα1 (green) homology model. Outlined is the linker motif (wherein the Y121C mutation resides) required for packing of GFRα1 domain 1 (D1; orange surface) against GFRα1 domain 3 (D3; grey surface). (**G**) Close-up view of the GFRα1 D1–D2 linker containing the highly conserved SPYE motif (human GFRα1 numbering). (**H**) The mutation in GFRα1 at position 121 substitutes the bulky polar side chain of tyrosine (Y; green sticks) with the shorter polar thiol group of cysteine (C; magenta sticks).

**Figure 2 genes-13-01687-f002:**
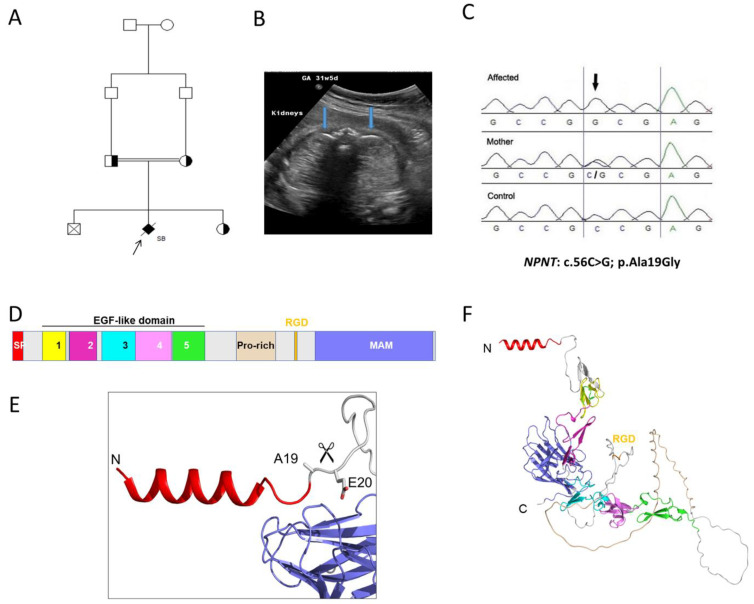
**Family 2 with renal cystic dysplasia and *NPNT* variant.** (**A**). Pedigree diagram of the studied family with cystic dysplasia with the proband arrowed. (**B**) Transverse view of the fetal abdomen at 31 weeks’ gestation, showing the fetal kidneys (blue arrows), which appear hyperechoic bilaterally. (**C**) Sanger sequencing chromatograms of the NPNT missense variant (NM_001033047.3:c.56C>G; p.Ala19Gly) in the proband, mother, and healthy control. (**D**) Domain structure of NPNT. NPNT contains an N-terminal signal peptide (SP), five EGF-like domains (Ca2+-binding by EGF-2, -4, and -5), compositional bias (Pro-rich), integrin interaction (RGD domain), meprin, A5/NRP1, and protein tyrosine phosphatase µ (MAM) domain 1. (**E**) AlphaFold2 predicted a three-dimensional structure of the N-terminal signal peptide domain. The SignalP 5.03–predicted cleavage site between Ala19 and Glu20 is indicated by the scissors. (**F**) Overall three-dimensional structure of NPNT predicted by AlphaFold2. Domains are colored as in D.

**Table 1 genes-13-01687-t001:** In silico tool predictions of detected missense variants in *GFRA1* and *NPNT*.

Gene	Transcript	Nucleotide Change	Amino Acid Change	PolyPhen-2	SIFT	FATHMM-MKL	DANN	REVEL	CADD Score	ACMG	CGM-DB (ƒ)	gnomAD (ƒ)
*GFRA1*	NM_001348098.4	c.362A>G	p.Tyr121Cys	Probably damaging (1.000)	Damaging (0.001)	Damaging (0.9356)	0.9985	Pathogenic (0.6309)	29.0	PM2, PP3	Variant not found	Variant not found
*NPNT*	NM_001033047.3	c.56C>G	p.Ala19Gly	Possibly damaging (0.841)	Tolerated (0.17)	Damaging (0.6652)	0.9944	Uncertain (0.324)	23.2	PM2	0.00042	0.0000107

PolyPhen-2: > 0.908 probably damaging, > 0.446 possibly damaging; SIFT < 0.05 deleterious; FATHMM-MKL: an integrated tool of noncoding and coding sequence variants; DANN: a deep learning tool for annotating the pathogenicity of variants; REVEL: rare exome variant ensemble learner, scores > 0.5 likely disease causing; CADD score: the Combined Annotation-Dependent Depletion tool, scores > 30 likely deleterious; ACMG: American College of Medical Genetics; CGM-DB: Centre for Genomic Medicine Database, Saudi Arabia; gnomAD: the genome aggregation database; ƒ: allele frequency.

## Data Availability

All data generated during this study are included in this published article.

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
