# Peer review of "Missense Variants in GFRA1 and NPNT Are Associated with Congenital Anomalies of the Kidney and Urinary Tract"

_genes, 2022, doi:10.3390/genes13101687_

Round 1

Reviewer 1 Report

Gist/Summary: The authors use cases of congenital anomalies of the kidney and urinary tract and exploit to udnerstand whether or not the missense mutations they are found are deleterious. 

While the strategy was to use whole exome sequencing-trio, they make a very little mention of results in the story and hurriedly put their emphasis on in silico modeling which is barely inviting with results. 

The results are half baked at either points:  NGS and modeling.  There is no strong rationale to circumvent the theory that the mutations they find would be at the interfacial sites.

Keeping in view of medical backgorund of authors, they could have largely exploited the reuslts of autosomal dominance/recessive in these patients and there is no points mentioned here. 

The manuscript also suffers from a good rationale and wekly knit small parahraphs all through. 

The ethics clearance and statement with samples collected ( antenatal) etc. neeeds clear mention  which I don't see.  Please correct me

The WES methods wer ejust written in one statement, how libraries were prepared, what amount of depth of coverage. herozygosity etc are not detailed. 

The figures also lack good visualization.A lot of merit could have been carrie dout frome xome to docking but the authors unfortunately fail to narrate a good story!

Sorry for not being positive!

Author Response

Thank you for the comments, please see a point-by-point response.

Referee 1

Gist/Summary: The authors use cases of congenital anomalies of the kidney and urinary tract and exploit to understand whether or not the missense mutations they are found are deleterious. 

While the strategy was to use whole exome sequencing-trio, they make a very little mention of results in the story and hurriedly put their emphasis on in silico modeling which is barely inviting with results. The results are half baked at either points:  NGS and modeling.  There is no strong rationale to circumvent the theory that the mutations they find would be at the interfacial sites.

Response: We have revised the manuscript to allow the story to be described more completely and in a more balanced way as suggested.

Keeping in view of medical background of authors, they could have largely exploited the results of autosomal dominance/recessive in these patients and there is no points mentioned here. 

Response: We have now added a discussion of the patterns of inheritance.

The manuscript also suffers from a good rationale and weakly knit small paragraphs all through. 

Response: We have proof read and revised the whole manuscript to improve the readability.

The ethics clearance and statement with samples collected ( antenatal) etc. needs clear mention  which I don't see.  Please correct me

Response: Thank you, we have added a statement to clarify the ethics.

The WES methods were just written in one statement, how libraries were prepared, what amount of depth of coverage. herozygosity etc are not detailed. 

Response: The additional details have now been added to the methods section.

The figures also lack good visualization. A lot of merit could have been carried out from exome to docking but the authors unfortunately fail to narrate a good story!

Response: We have tried to narrate a clearer story within the appear to improve the manuscript. We have provided high resolution figures for the revised version.

Reviewer 2 Report

At the outset, I would like to congratulate the authors for their work. The study has merit and will be of interest to our readers. However, I have some comments:

Introduction: Please add your hypothesis at the end of this section. What did you hypothesize before conducting this research?

Methods: Please specify how the samples were collected from fetuses.

-For family 2, please mention how many family members were enrolled in the genetic testing.

-Were healthy (and unrelated) controls also included? What's the frequency of this mutation in controls?

-What are the other associations of this mutation?

-Did the NGS identify mutations in other genes? Were other genes also studied in this study?

Discussion: Please add a paragraph on limitations and future perspectives.

Author Response

Thank you for the comments, please see a point-by-point response.

Referee 2

At the outset, I would like to congratulate the authors for their work. The study has merit and will be of interest to our readers. However, I have some comments:

Introduction: Please add your hypothesis at the end of this section. What did you hypothesize before conducting this research?

Response: Thank you for the suggestion. We have now added hypothesis sections to the introduction.

Methods: Please specify how the samples were collected from fetuses.

Response: Thank you, we can clarify that fetal DNA was extracted cord blood.

-For family 2, please mention how many family members were enrolled in the genetic testing.

Response: For the discovery cohort, fetal DNA combined with both parents were used for Trio-WES. Subsequently the younger sibling was tested for the familial variant using Sanger sequencing. We have clarified the text and added methods for Sanger sequencing.

-Were healthy (and unrelated) controls also included? What's the frequency of this mutation in controls?

Response: For family1, 5  asymptomatic siblings were included and were either wild type or heterozygous for the GFRA1 allele. This variant is novel and was not found in gnomAD or our in house database (Centre for Genomic Medicine Database, Saudi Arabia CCGM-DB). Table 1 contains this information.

For family 2, trio-WES was performed using the fetal DNA and both parents. Subsequently the younger sibling was found to be heterozygous for the NPNT allele following Sanger sequencing. The in house (Centre for Genomic Medicine Database, Saudi Arabia CCGM-DB) allele frequency was 0.0042 and the gnomAD frequency was 0.0000107. Table 1 contains this information.

-What are the other associations of this mutation?

Response: The NPNT gene is so far only associated with bilateral renal agenesis. It is possible that hypomorphic variants may give rise to milder CAKUT phenotypes.

-Did the NGS identify mutations in other genes? Were other genes also studied in this study?

Response: We performed whole exome sequencing. Following our data pipeline, there were no other plausible variants identified that would account for the phenotype.

Discussion: Please add a paragraph on limitations and future perspectives.

Response: We have added an additional discussion paragraph as suggested.

Reviewer 3 Report

In this manuscript, the authors identified novel missense mutations to GFRA1 and NPNT that might cause congenital anomalies of the kidney and urinary tract (CAKUT). Both GFRA1 and NPNT have well established roles in kidney development and loss-of-function mutations of both genes have recently been reported to be linked to CAKUT. This study, however, still represents the first report of missense mutations in these two genes as potential causality of CAKUT. Overall, the study is original, and the findings are important to our understanding of human CAKUT. One minor suggestion to improve the readability: Although the authors (and likely researchers in closely relevant research areas) are very familiar with CMA testing and ES, it would be necessary to describe the full names of CMA and ES, the first time these terms were used. Considering they are important assays in this study, it would be better if some sentences can be added to briefly describe these assays as well.

Author Response

Thank you for the comments, please see a point-by-point response.

Referee 3

In this manuscript, the authors identified novel missense mutations to GFRA1 and NPNT that might cause congenital anomalies of the kidney and urinary tract (CAKUT). Both GFRA1 and NPNT have well established roles in kidney development and loss-of-function mutations of both genes have recently been reported to be linked to CAKUT. This study, however, still represents the first report of missense mutations in these two genes as potential causality of CAKUT. Overall, the study is original, and the findings are important to our understanding of human CAKUT.

One minor suggestion to improve the readability: Although the authors (and likely researchers in closely relevant research areas) are very familiar with CMA testing and ES, it would be necessary to describe the full names of CMA and ES, the first time these terms were used. Considering they are important assays in this study, it would be better if some sentences can be added to briefly describe these assays as well.

Response: Thank you for the comments, we have added some descriptions of CMA and ES to help understanding and readability of the manuscript.

Round 2

Reviewer 1 Report

I could see the mmanuscript greatly improved.  May I suggest the authors to add conclusions and subhead and add few more sentences?

Author Response

Reviewer-1 (2nd round):

I could see the manuscript greatly improved.  May I suggest the authors to add conclusions and subhead and add few more sentences?

Response:

Thank you. We have added some sub headings and expanded the conclusions as suggested.